# Cost-effectiveness of alternative minimum recall intervals between whole blood donations

Zia Sadique[1]*, Sarah Willis[1], Kaat De Corte[1], Mark Pennington[1,2], Carmel Moore[3,4], Stephen Kaptoge[3], Emanuele Di Angelantonio[3], Gail Miflin[5], David J. Roberts[3,5,6], Richard Grieve[1]

1 Department of Health Services Research and Policy, London School of Hygiene and Tropical Medicine, London, United Kingdom, 2 Department of Health Services and Population Research, King's College London, London, United Kingdom, 3 Department of Public Health and Primary Care, NIHR Blood and Transplant Research Unit in Donor Health and Genomics, University of Cambridge, Cambridge, United Kingdom, 4 University of East Anglia, Norwich, United Kingdom, 5 NHS Blood and Transplant, Liverpool, United Kingdom, 6 Radcliffe Department of Medicine and BRC Biomedical Centre–Haematology Theme, University of Oxford, John Radcliffe Hospital, Oxford, United Kingdom

* zia.sadique@lshtm.ac.uk

**Data Availability Statement:** All relevant data are within the manuscript and its Supporting Information files.

## Abstract

### Background

The INTERVAL trial showed shorter inter-donation intervals could safely increase the frequency of whole-blood donation. We extended the INTERVAL trial to consider the relative cost-effectiveness of reduced inter-donation intervals.

### Methods

Our within-trial cost-effectiveness analysis (CEA) used data from 44,863 whole-blood donors randomly assigned to 12, 10 or 8 week (males), and 16, 14 or 12 week inter-donation intervals (females). The CEA analysed the number of whole-blood donations, deferrals including low-haemoglobin deferrals, and donors' health-related quality of life (QoL) to report costs and cost-effectiveness over two years.

### Findings

The mean number of blood donation visits over two years was higher for the reduced interval strategies, for males (7.76, 6.60 and 5.68 average donations in the 8-, 10- and 12-week arms) and for females (5.10, 4.60 and 4.01 donations in the 12-, 14- and 16-week arms). For males, the average rate of deferral for low haemoglobin per session attended, was 5.71% (8-week arm), 3.73% (10-week), and 2.55% (12-week), and for females the rates were: 7.92% (12-week), 6.63% (14-week), and 5.05% (16-week). Donors' QoL was similar across strategies, although self-reported symptoms were increased with shorter donation intervals. The shorter interval strategies increased average cost, with incremental cost-effectiveness ratios of £9.51 (95% CI 9.33 to 9.69) per additional whole-blood donation for

**Funding:** The authors have disclosed the following funding: NIHR Health Services and Delivery Research Programme (GB), HS&DR – 13/54/62.

**Competing interests:** The authors have declared that no competing interests exist.

the 8- versus 12- week interval for males, and £10.17 (95% CI 9.80 to 10.54) for the 12- versus 16- week interval arm for females.

## Conclusions

Over two years, reducing the minimum donation interval could provide additional units of whole-blood at a small additional cost, including for those donor subgroups whose blood type is in relatively high demand. However, the significance of self-reported symptoms needs to be investigated further before these policies are expanded.

## Introduction

The safe and adequate supply of blood is an integral part of any health system. All health systems share a global vision for a self-sufficient supply of whole-blood by 2020, as set out by the World Health Organization (WHO) and the International Federation of Red Cross and Red Crescent Societies (IFRC) [1]. This framework for global action focuses on the importance of voluntary blood donors for blood safety and availability and called for blood supply agencies to encourage more frequent donation from current whole-blood donors. In recent years the demand for whole-blood has declined overall in many high-income countries, but the demand for universal blood type and some rare subtypes has been growing. In England, there is increased demand for the universal blood type O negative (O-),A negative (A-), B negative (B-) and rare blood subtypes (Ro subtypes) more common in black, Asian and minority ethnic (BAME) donors and supply of these blood types is particularly vulnerable to shortfalls. Further threats to the sustainability of voluntary whole-blood services in England are the gender gap in recruiting new donors and difficulty in retaining younger blood donors [2–4].

NHSBT's blood collection service has been severely affected by the ongoing COVID-19 pandemic. In March 2020 the level of donation was 15% lower than expected [5]. The fall in supply has been mitigated by the cancellation of elective procedures, but raises an important challenge for ensuring that the supply of whole-blood is sufficient in the post-COVID-19 recovery period when demand for blood will be high. A key policy objective of NHSBT is to collect more blood in particular for blood types that are relatively in high demand. So evidence to inform changes to the blood service that increase donation frequency for subgroups of donors whose whole-blood type is in high demand, at low additional cost is timely and potentially of strategic importance. However, rigorous evidence about the effects of changes to the blood collection service on the frequency and costs of whole-blood donation is lacking, with most existing economic evaluations based on non-randomised evidence [6–10].

INTERVAL is the first ever randomised controlled trial (RCT) that investigated the efficiency and safety of alternative blood donation services. The INTERVAL RCT assessed whether reducing inter-donation intervals in static donor centres of NHSBT in England would increase donation frequency without compromising donor safety [11]. The trial reported that for both genders, donors randomised to the shorter minimum donation interval (8 weeks for men, 10 weeks for women) successfully donated more whole-blood over two years compared to those randomised to the current minimum donation intervals (12 weeks for men, 16 weeks for women). However, even after the prescribed inter-donation intervals, some donors may fail to regain their previous haemoglobin concentration and fail to pass the haemoglobin threshold mandated for donation (135 g/L for men and 125g/L for women). More frequent donations (ie, shorter inter-donation intervals) were associated with higher rates of deferral

for low haemoglobin over two years follow-up period in the trial. The subsequent extension to the INTERVAL trial that followed donors up for four years, and offered routine rather than intensive reminders, also found that shorter inter-donation intervals continued to increase donation frequency but increased deferral rates [12].

Neither of these reports of the INTERVAL trial considered the relative costs of reduced inter-donation intervals which could be higher given the additional deferrals, nor did they evaluate the effects for policy-relevant subgroups, such as those donors whose blood is in 'high demand' (for example, O negative (O-), A negative (A-), B negative (B-) and blood subtypes more common in black, Asian and minority ethnic (BAME) donors or those donor subgroups who are less likely to continue donating (younger donors, or those who have made relatively few previous donations).

The Health Economics Modelling of alternative blood donation strategies (HEMO) study set out to assess the cost-effectiveness of strategies to maintain the blood supply in England [13]. This paper reports findings from the cost-effectiveness analysis (CEA) of the alternative minimum inter-donation intervals considered over two-years within the INTERVAL trial. This paper extends the CEA published in the NIHR report, in providing a comprehensive assessment of the relative cost-effectiveness of alternative inter donation intervals according to pre-specified policy relevant subgroups for both genders. The subgroups of interest are blood type, age, ethnicity, donor recruitment source and whether the donor was giving blood for the first time or a regular donor.

## Methods

### Ethics

The INTERVAL trial protocol was approved by the National Research Ethics Service (11/EE/0538). The trial was registered with the International Standard Randomised Controlled Trial Number (ISRCTN) Registry (ISRCTN24760606).

### Setting, selection and baseline measures

The INTERVAL was an open, parallel-group pragmatic RCT that recruited whole blood donors aged 18 years or older from 25 static donor centres of NHSBT across England [11,12,14,15]. The initial findings from the INTERVAL trial and the study protocol, are reported elsewhere [11,12,14,15]. In brief, new and existing donors were eligible for inclusion in the trial if they were: aged 18 years or older, met the routine criteria for whole blood donation, were willing to be randomised, had an email address and access to the internet to respond to web-based questionnaires, and were willing to be randomly assigned to any of the trial's intervention groups at one of the 25 static donor centres of NHSBT. Existing donors were defined as donors who had given blood within the last five years. Written consent was obtained from eligible donors, who were asked to complete and sign two copies of the consent form. Completed consent forms were checked for completion of all relevant sections and for the donor's signature. The 'study copy' of the consent form, affirmed by signature by a staff member of the study that he/she had witnessed its completion was retained while the 'donor copy' was provided to the participant. For donors who subsequently were ineligible or unwilling to take part in the trial, consent forms were crossed through and then destroyed. Male participants were randomly assigned to 12- versus 10- versus 8-week inter-donation intervals and female participants were randomly assigned to 16- versus 14- versus 12-week inter-donation intervals. Those donors who were eligible and consented, were randomised to the three gender-specific intervention groups in a 1:1:1 ratio.

This study excluded those donors who withdrew consent, who died during or after the trial follow-up period until December 2016 when linked PULSE (the NHSBT national blood supply database) data were extracted, or who did not have requisite PULSE data available. This led to an overall sample of 44,863 trial participants for the cost-effectiveness analysis. The follow-up period of the study was two years.

Information for baseline characteristics (gender, age, ethnicity and blood type) and donation history (new donor or not, recruitment source, and the number of donations and deferrals for low haemoglobin (Hb) for the two years prior to randomisation) of trial participants was extracted from PULSE database. At the baseline donation visit after trial recruitment, a full blood count was performed which provided the levels of Hb used to define the proportion of low Hb deferrals who would require additional consultations and tests. Trial participants were asked to complete a baseline questionnaire online, which included the SF-36 (Short Form 36) questionnaire [16].

## Resource use and costs

The cost analysis took a NHS and personal social services perspective as recommended by the National Institute for Health and Care Excellence (NICE) [17]. The study included cost items that were anticipated to differ over the trial follow-up period and according to intervention groups and included the additional costs of blood collection excluding processing, marketing or fixed costs, cost of deferrals and subsequent health care costs. The relevant additional staff costs, costs of invitation and consumables costs associated with blood collection were included in the study.

The number of successful whole blood donations, deferrals and fainting episodes at a blood donation session were extracted from the PULSE database over the two-year follow-up period. The volume of blood donated was measured in units of whole blood (each unit is 470ml). Donations could be deferred for a number of reasons, such as recent travel, medication, lifestyle restrictions or infection/illness. as described in the donor selection guidelines (https://www.transfusionguidelines.org/dsg). Donors could also be deferred due to low Hb, which was anticipated to differ by randomised arm. The trial used the same deferral policy that is used in routine practice as per the Blood Safety Quality Regulations, for example, donors with Hb levels that were 'low', that is less than 135g/L for males and 125g/L for females, were deferred for three months. All deferrals were associated with resource use consequences in terms of staff time, Hb screening test and downstream healthcare costs (GP appointment, full blood count test, ferritin test, iron supplement, and hospital outpatient appointment) in the case of Hb-related deferrals.

Web-based follow-up questionnaires collected information on health care events occurring between donation sessions (doctor or hospital visits required for falls, transport accidents, angina, heart failure, transient ischaemic attack, stroke, myocardial infarction). While the numbers of these events were reported, they were not anticipated to differ between the randomised arms, and were not included in the cost analysis. Fainting event at donation sessions were anticipated to differ between randomised groups and was included in the cost analysis.

Unit costs were taken from NHSBT financial records, expert opinion, and INTERVAL trial data (see Appendix Table 1 in S1 Appendix). The unit cost of donation appointment reminders was calculated, according to the three-stage reminder process (first appointment, interim appointment and last appointment reminders) specified by the INTERVAL trial protocols. Time required for sending the reminders recorded in the trial were costed according to NHS Band 4 costs [18]. The opportunity cost of staff time lost following a donor deferral whether due to low haemoglobin or other reasons was based on expert opinion. The major opportunity

cost of an additional deferral is the reduced efficiency of collection, that is the number of units of blood collected by a team during a donation session. The opportunity cost therefore includes the time taken for donor carers (NHS Band 4) to undertake a health screen and, where deferral was due to low Hb, a copper sulphate [19] and HemoCue® test (HemoCue®, Radiometer Medical ApS, Denmark) [20]. Informed by INTERVAL trial data we assumed that 7% of donors with low Hb would be referred to their Primary Care Physician (when Hb is less than 125 g/L for men and 115 g/L for women) which would incur healthcare costs. The health-care costs associated with low Hb were assumed to include a GP appointment, a full blood count test and Serum ferritin test, iron supplements (in 50% of cases) and an outpatient appointment (in 10% cases).

The accompanying unit costs were taken from published sources [18,21–25]. The unit cost of a fainting episode was calculated according to the additional staff (NHS Band 4) time required at a donor centre to manage a typical fainting episode. The unit costs related to blood collection were taken from NHSBT financial records [19]. Resource use data were combined with unit costs to report the total costs for each randomised donor over the trial's two-year fol-low-up period.

## Health outcomes

The main health outcomes for the cost-effectiveness analysis were successful whole blood donations, overall donation deferrals, donation deferrals due to low Hb, and quality of life (QoL). Whole blood donations, donation deferrals due to low Hb and donation deferrals due to other causes were recorded in the trial database. Participants were sent a request by email to complete an online questionnaire, which included the SF-12 (Short Form 12), at six, 12- and 18-months follow-up, and the SF-36 at the final two year follow-up timepoint. The responses to the required SF-12 & SF-36 questions were combined with the published valuation algo-rithm [26] to report SF-6D (Short-Form Six-Dimension) utility score at each timepoint, anchored on the scale 0 (death) and 1 (perfect health).

## Cost-effectiveness analysis

The cost-effectiveness analysis followed the intention-to-treat principle [27]. The time horizon was two years, as per the follow-up period of the INTERVAL trial. The analysis applied logistic regression models for estimating deferral rates, Generalised Estimating Equation (GEE) mod-els for estimating SF-6D score, and seemingly-unrelated regression (SUR) for joint modelling of whole blood donations and cost [28]. Rates of deferral was estimated using the data on num-ber of deferrals and attendances, and applied logistic regression models for grouped data. SF-6D score at each time point of measurement (baseline, six month, 12-month, 18-month and 24-month) was estimated using GEE model. Costs and whole blood donations were estimated jointly by applying a SUR model that accounted for the correlation between whole blood dona-tion and costs.

The cost-effectiveness analysis adjusted for age, 'standard' (donors with blood types O+, A +, B+, AB+ and AB) versus 'high' (donors with blood types O-, A- and B-) demand blood types, ethnicity (white, Asian/Asian mixed, Black/Black mixed, other ethnicity or not stated), new donor or not, and recruitment source (static donor centre vs. mobile session vs. other). Subgroup effects were estimated by including interaction terms for randomised arm by sub-group. Age was defined as a continuous variable in the analysis model, but predictions were provided according to the requisite policy-relevant categories (17–30, 31–45, 46–60, 60+).

QoL data was missing for those individuals who did not complete the items required to cal-culate the SF-6D utility score; the number and percentage of the analysis sample with

responses sufficient to calculate the SF-6D utility score are reported for each timepoint (baseline, six, 12, 18 and 24 months) (see Appendix Table 2 in S1 Appendix). These missing data were handled by a GEE model that included SF-6D utility score as the dependent variable, with randomised group, timepoint, and the above subgroup variables as the fixed effects of interest, together with fixed interaction terms of timepoint and randomised group. The model included random intercepts for donor centre and individual, to allow for the correlations of measurements within each donor and donor centre. The model reported mean QoL scores at each timepoint within the two-year follow-up of the trial, and the differences in the mean utility scores across the randomised arms. The GEE model assumed that missing QoL data were 'missing at random', conditional on the variables included in the model [29].

We reported incremental (difference in means) costs and number of whole blood donations, and the incremental cost-effectiveness ratio (ICER), as the incremental cost per additional unit of blood donated by those allocated to the reduced inter-donation intervals compared to those giving blood at the standard interval for men and women respectively. The confidence intervals around the ICER were constructed by applying Delta method (Taylor series expansion on the incremental estimates of cost and volume of blood donated) [30]. The accompanying uncertainty around the incremental estimates of cost and the volume of blood donated was represented on the cost-effectiveness plane. We report results overall (by gender), and according to the other pre-specified subgroups.

The base case analyses assumed unit costs for reminders to donate and deferrals from expert opinion; zero costs for non-attendances; downstream health care costs following a deferral due to low Hb; and costs attributable to fainting episodes. We also assumed that static donor centres had staff capacity to collect more bloods. The statistical models for blood volume, QoL and cost assumed that the residuals follow a Normal distribution. The robustness of the results to these assumptions was assessed in the subsequent sensitivity analyses.

## Results

### Patient characteristics

The baseline characteristics (Table 1) were similar between the randomised groups for both genders. The number of blood donations, deferral for low Hb and for reasons other than low Hb in the two years preceding the trial and baseline QoL were also similar across randomised groups for both genders.

### Resource use and costs

The resource use results reported in Table 2 shows that mean number of blood donation visits was relatively higher in reduced minimum donation interval arms for both genders. For males, the mean number of blood donation visits was 7.76 in the 8-week arm, compared to 6.60 and 5.68 in the 10- and 12- week arms. For females the corresponding mean number of blood donation visits was 5.10 in the 12-week arm, compared to 4.60 and 4.01 in the 14- and 16- week arms. The average rate of deferral for low Hb, per session attended, was higher in reduced minimum donation intervals arms for both genders (see Table 2, Appendix Table 3A & 3B in S1 Appendix). For males, Hb-related deferral rate was 5.71% in the 8- week arm, which was relatively higher compared to 3.73% in the 10-, and 2.55% in the 12- week arm. For females, this deferral rate was 5.05% in the 16- week arm compared to 6.63% in the 14-, and 7.92% in the 12- week arm. In accordance with the rate of deferrals the mean number of Hb-related deferrals per donor over two years were also higher in the randomised arms with reduced inter-donation intervals. While the rates and mean number of Hb-related deferrals were higher for randomised groups with reduced inter-donation intervals, the proportion of

**Table 1. Baseline characteristics, by randomised arm and gender.**

| | | Randomised arm (male) | | | Randomised arm (female) | | |
|---|---|---|---|---|---|---|---|
| | | 8-week (n = 7,417) | 10-week (n = 7,413) | 12-week (n = 7,411) | 12-week (n = 7,549) | 14-week (n = 7,545) | 16-week (n = 7,528) |
| **Mean (SD) age (years)** | | 44.7 (14.1) | 44.7 (14.2) | 44.7 (14.2) | 40.77 (14.0) | 40.89 (13.9) | 40.94 (14.0) |
| **Blood type n (%)** | High demand | 996 (13.43) | 933 (12.59) | 965 (13.02) | 1,130 (14.97) | 1,062 (14.08) | 1,002 (13.31) |
| | Standard demand | 6,421 (86.57) | 6,480 (87.41) | 6,446 (86.98) | 6,419 (85.03) | 6,483 (85.92) | 6,526 (86.69) |
| **Ethnicity n (%)** | White | 6,751 (91.02) | 6,752 (91.08) | 6,745 (91.01) | 6,984 (92.52) | 6,992 (92.67) | 6,949 (92.31) |
| | Black/mixed black | 101 (1.36) | 96 (1.30) | 100 (1.35) | 103 (1.36) | 93 (1.23) | 134 (1.78) |
| | Asian/mixed Asian | 255 (3.44) | 271 (3.66) | 258 (3.48) | 171 (2.27) | 177 (2.35) | 154 (2.05) |
| | Other or not stated | 310 (4.18) | 294 (3.97) | 308 (4.16) | 291 (3.85) | 283 (3.75) | 291 (3.87) |
| **New donor n (%)** | No | 6,817 (91.91) | 6,818 (91.97) | 6,818 (92.00) | 6,742 (89.31) | 6,744 (89.38) | 6,727 (89.36) |
| | Yes | 600 (8.09) | 595 (8.03) | 593 (8.00) | 807 (10.69) | 801 (10.62) | 801 (10.64) |
| **Recruitment source n (%)** | Centre | 4,907 (66.16) | 4,840 (65.29) | 4,855 (65.51) | 4,851 (64.26) | 4,921 (65.22) | 4,901 (65.10) |
| | Mobile | 1,437 (19.37) | 1,510 (20.37) | 1,512 (20.40) | 1,545 (20.47) | 1,482 (19.64) | 1,486 (19.74) |
| | No invite | 1,073 (14.47) | 1,063 (14.34) | 1,044 (14.09) | 1,153 (15.27) | 1,142 (15.14) | 1,141 (15.16) |
| **Mean (SD) deferrals for low Hb in previous 2 years** | | 0.04 (0.24) | 0.04 (0.23) | 0.04 (0.24) | 0.12 (0.39) | 0.12 (0.38) | 0.12 (0.39) |
| **Mean (SD) deferrals for other reasons in previous 2 years** | | 0.32 (0.69) | 0.32 (0.68) | 0.32 (0.69) | 0.36 (0.68) | 0.34 (0.68) | 0.34 (0.68) |
| **Mean (SD) number of blood donation visits in previous 2 years** | | 4.19 (2.40) | 4.22 (2.42) | 4.18 (2.40) | 3.46 (1.91) | 3.45 (1.89) | 3.44 (1.93) |
| **Mean (SD) SF-6D score at baseline** | | 0.86 (0.08) | 0.86 (0.08) | 0.86 (0.09) | 0.85 (0.09) | 0.85 (0.09) | 0.85 (0.09) |

deferrals due to other reasons, mean fainting episodes, and other donor-reported health care events (Table 2 & Appendix Table 4 in S1 Appendix), were similar across the randomised groups for both genders.

The total mean costs per male donor at two-years were relatively lower for reduced minimum donation interval arm for both genders. The corresponding mean costs for males were £61, £52 and £45 in the 8-, 10- and 12- week arms. The mean costs for females were £41, £37 and £33 in the 12-, 14- and 16- week arms (Table 3).

## Health outcomes

The estimated effects of randomised group on the mean SF-6D scores at each timepoint are reported in Appendix Table 5A & 5B in S1 Appendix. There was no difference in QoL (SF-6D score) between the randomised groups, at the two-year follow-up (Table 3), and at each of the

**Table 2. Resource use over two-year follow-up period, by randomised arm and gender.**

| | Randomised arm (male) | | | Randomised arm (female) | | |
|---|---|---|---|---|---|---|
| | 8-week (n = 7,417) | 10-week (n = 7,413) | 12-week (n = 7,411) | 12-week (n = 7,549) | 14-week (n = 7,545) | 16-week (n = 7,528) |
| **Mean blood donations visits** | 7.76 | 6.60 | 5.68 | 5.10 | 4.60 | 4.01 |
| **Deferrals for low Hb per attendance (%)** | 5.71 | 3.73 | 2.55 | 7.92 | 6.63 | 5.05 |
| **Deferrals for other reasons per attendance (%)** | 4.36 | 4.58 | 4.79 | 6.57 | 6.95 | 7.28 |
| **Mean deferrals for low Hb per donor** | 0.44 | 0.25 | 0.15 | 0.40 | 0.30 | 0.20 |
| **Mean deferrals for other reasons per donor** | 0.33 | 0.30 | 0.27 | 0.34 | 0.32 | 0.29 |
| **Mean faints per donor** | 0.02 | 0.02 | 0.02 | 0.04 | 0.03 | 0.03 |

**Table 3. SF-6D score (at two years), whole blood donations, costs and incremental cost per additional unit of whole blood donated, over two-year follow-up (by gender).**

| | Male | | | | | Female | | | | |
|---|---|---|---|---|---|---|---|---|---|---|
| | Randomised arm | | | Mean (95% CI) difference | | Randomised arm | | | Mean (95% CI) difference | |
| | 8-week (n = 7,417) | 10-week (n = 7,413) | 12-week (n = 7,411) | 8-week vs. 12-week | 10-week vs. 12-week | 12-week (n = 7,549) | 14-week (n = 7,545) | 16-week (n = 7,528) | 12-week vs. 16-week | 14-week vs. 16-week |
| **Mean SF-6D score** | 0.84 | 0.84 | 0.84 | 0.002 (-0.002 to 0.006) | -0.001 (-0.004 to 0.003) | 0.82 | 0.82 | 0.82 | 0.001 (-0.003 to 0.005) | 0.003 (-0.001 to 0.007) |
| **Mean whole blood donations[±]** | 6.89 | 5.98 | 5.19 | 1.71 (1.60 to 1.80) | 0.79 (0.70 to 0.88) | 4.29 | 3.91 | 3.45 | 0.85 (0.78 to 0.92) | 0.46 (0.40 to 0.53) |
| **Mean costs (£)[±]** | 61 | 52 | 45 | 16 (15 to 17) | 7 (6 to 8) | 41 | 37 | 33 | 9 (8 to 9) | 5 (4 to 5) |
| **Incremental cost-effectiveness ratio[±]** | | | | 9.51 (9.33 to 9.69) | 9.00 (8.66 to 9.34) | | | | 10.17 (9.80 to 10.54) | 9.98 (9.32 to 10.64) |

[±] The results for whole blood donations are rounded to two decimal places and costs are rounded to no decimal place. The incremental cost-effectiveness ratio results are rounded to two decimal places.

intervening time-points (Appendix Table 6 in S1 Appendix) between people who gave blood most and least frequently.

## Cost-effectiveness

The cost-effectiveness results are summarised in Table 3 and the regression coefficients from the joint estimation of costs and number of whole blood donations are reported Appendix Table 7 in S1 Appendix. For both genders, the average QoL score were similar between the randomised groups. The differences in mean QoL between randomised groups were small but the 95% CI included zero. Reduced inter-donation interval strategies were associated with higher number of donations. For males, compared to 12-week randomised group (who gave blood least frequently) the average number of whole blood donations over the two years follow-up period increased by 1.71 (95% 1.60 to 1.80) for the 8- week arm, and by 0.79 (95% CI from 0.70 to 0.88) for the 10- week arm. For females the corresponding increase in the average number of donations over the two years follow-up period was 0.85 (95% CI from 0.78 to 0.92) for 12- versus 16 weeks, and 0.46 (95% CI 0.40 to 0.53) for 14- versus 16- weeks. The reduced inter-donation interval strategies were also associated with higher costs. The corresponding ICERs were £9.51 (95% CI 9.33 to 9.69) for the 8-versus 12-week interval arm for males, and £10.17 (95% CI 9.80 to 10.54) for the 12-versus 16-week interval arm for females. The distributions of the mean costs and mean number of donations plotted on the cost-effectiveness plane shows that the joint distribution of costs and number of donations are centred tightly around the means (see Appendix Figure 1 in S1 Appendix).

The subgroup results show that including interaction effects for subgroups by randomised group improved model fit and the interaction term was statistically significant (for males the likelihood test results reported $chi^2$ = 79.28, p = 0.0002; for females, $chi^2$ = 46.55, p = 0.0153). However, the subgroup results (Figs 1–3) show that the incremental cost-effectiveness results were generally similar across almost all subgroups, albeit with considerable uncertainty surrounding the results. The level of uncertainty is higher for the ethnicity subgroup, especially for Asian/mixed Asian and black/mixed black ethnicity where mean incremental costs, whole blood donations, and ICERs have wide confidence intervals for both genders. For the comparison of 14- versus 16-week minimum donation interval strategies for women whose ethnicity

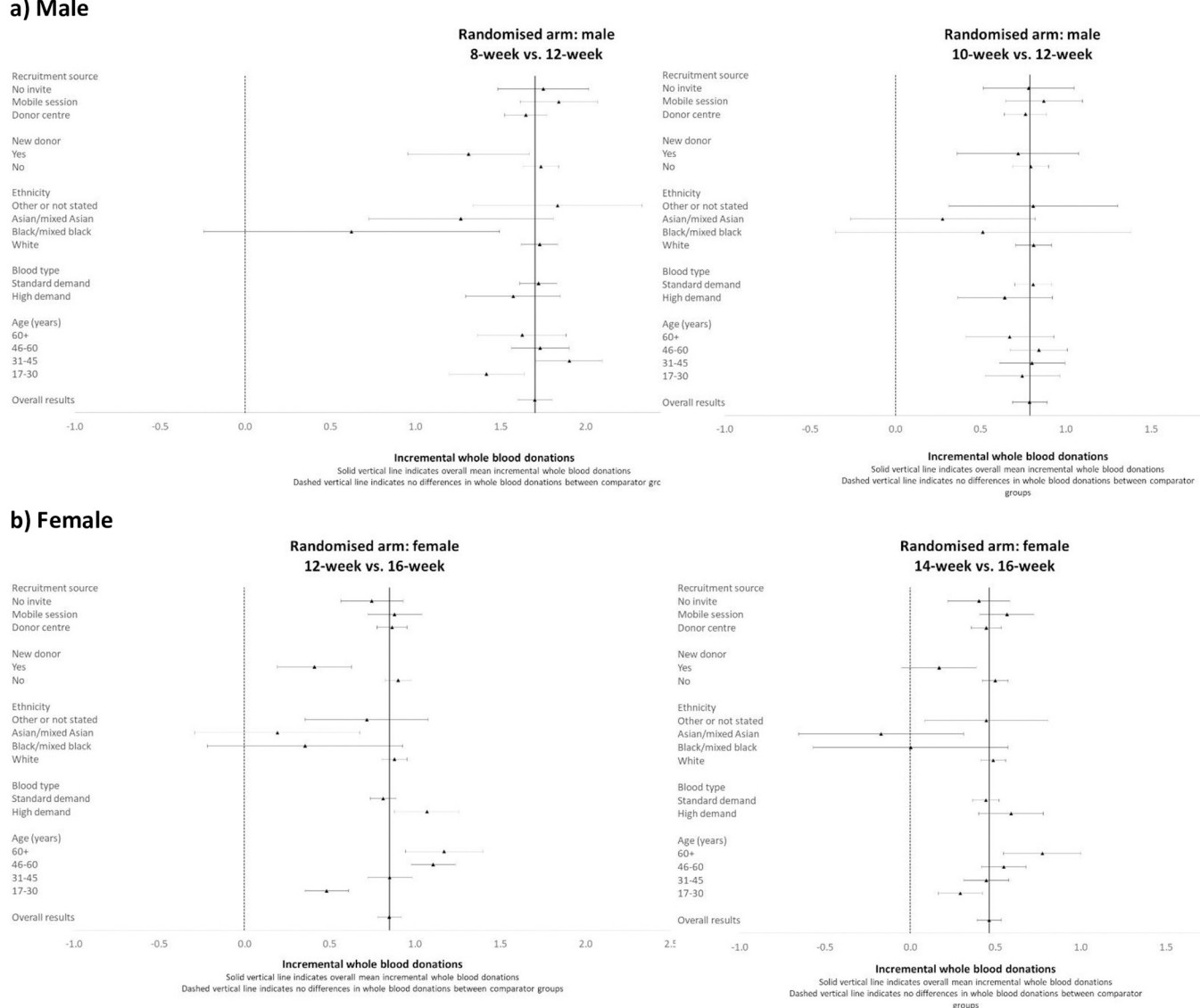

**Fig 1. Mean (95% CI) incremental blood donations over two-year follow-up period by subgroup.** a) Male b) Female.

was defined as black/mixed black, the incremental effect of the reduced interval on the number of whole blood donations was small, and so the accompanying mean ICER was large (£258). However, the sample size for this subgroup is low (n = 330 across all 3 arms), and the estimated ICERs are somewhat uncertain.

For females, whose blood type is in 'high demand', and for older women, the strategies of reduced inter-donation intervals led to a greater average increase in donation frequency than for donors whose blood type was in 'standard demand' and younger women. Hence the estimated ICERs were somewhat lower than for women whose blood type is in 'high demand' and older age groups.

The subgroup results for new versus experienced donors reported similar ICERs. The sensitivity analyses show that the base case cost-effectiveness results were generally not sensitive to alternative assumptions considered in the cost-effectiveness analysis (Fig 4). The base case

**a) Male**

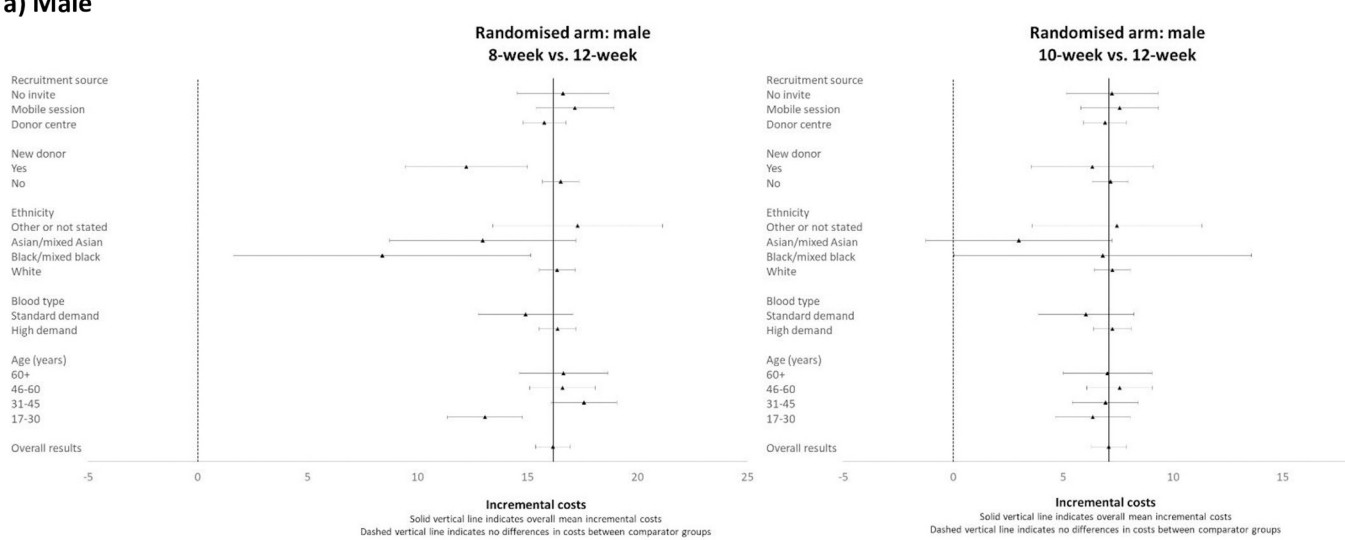

**b) Female**

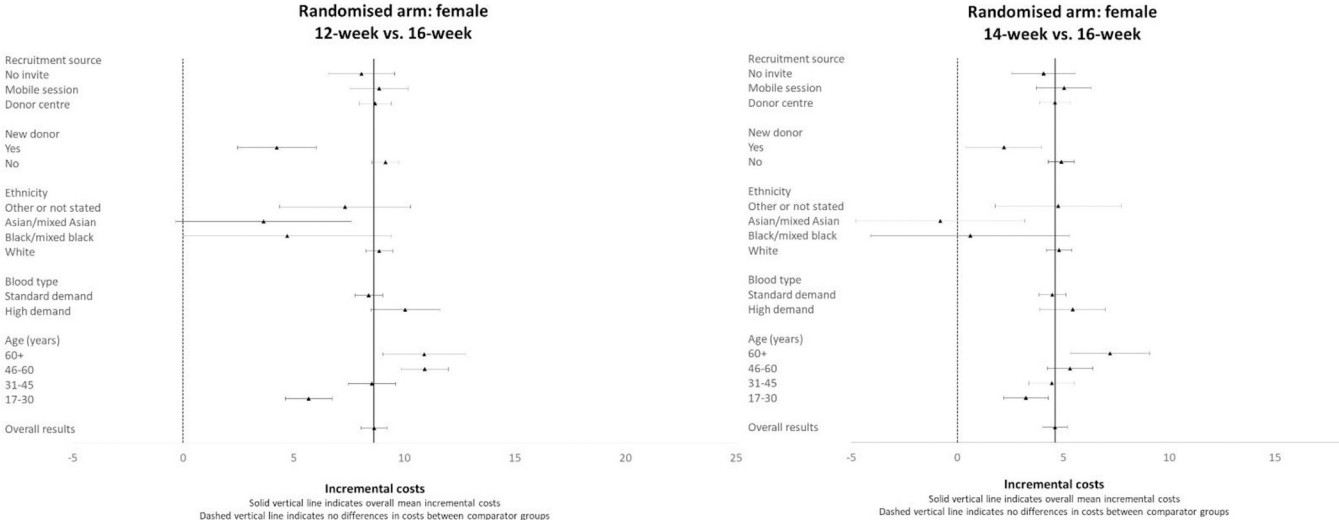

**Fig 2. Mean (95% CI) incremental costs over two-year follow-up period by subgroup.** a) Male b) Female.

results for both males and females were most sensitive to the inclusion of the additional staff costs required to collect extra blood followed by additional cost of non-attendance and excluding healthcare costs due to Hb deferral and alternative distributional assumption for costs. The base case results were not sensitive to the exclusion of invitation and fainting costs.

## Discussion

Our study is the first ever cost-effectiveness analysis of different inter-donation interval strategies and uses data from a large trial in real life setting. We find that reduced minimum donation interval strategies increase the average number of donations, at a small additional average cost over two years. The study finds that frequent blood donation is more cost-effective for those females whose blood group is in 'high demand' and for older female donors. For all other subgroups the cost-effectiveness results are similar. The study also finds that the rate of

## a) Male

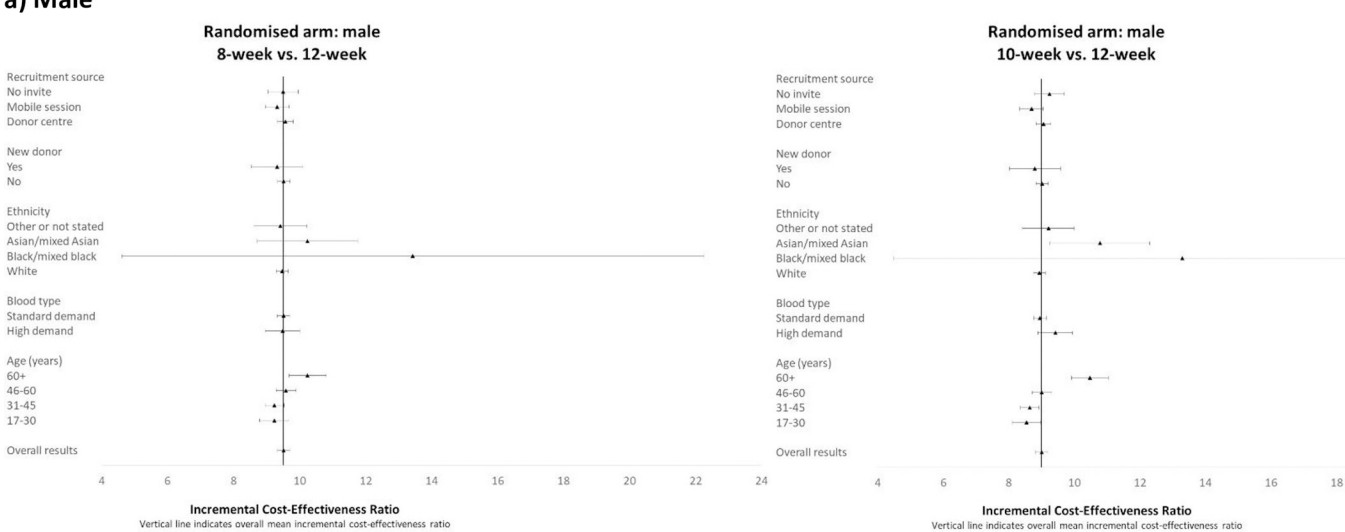

## b) Female

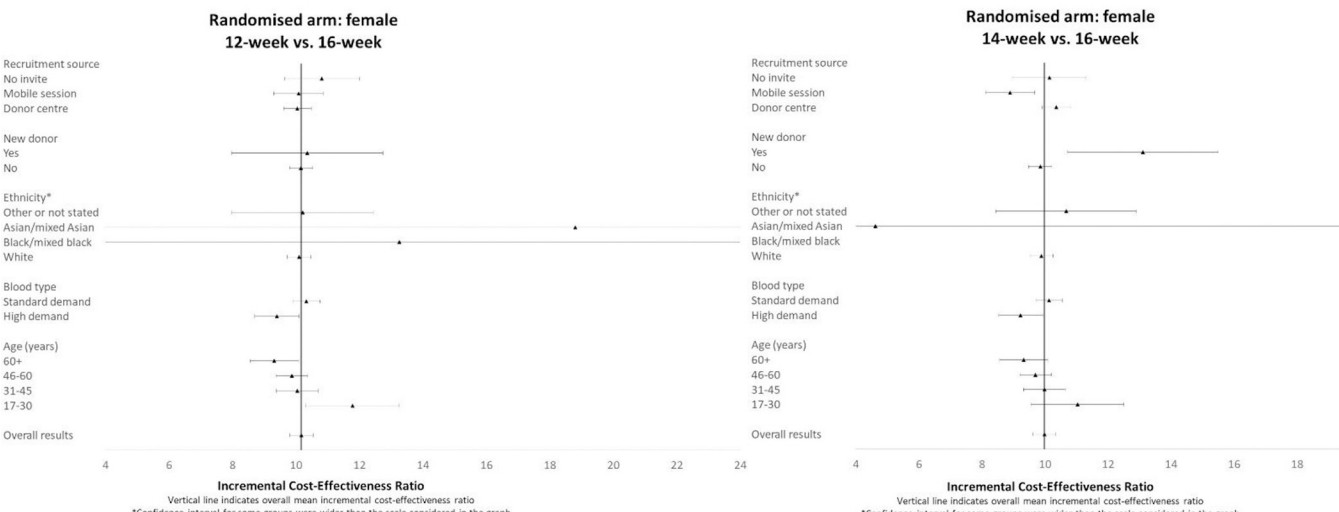

**Fig 3. Mean (95% CI) incremental cost-effectiveness ratios over two-year follow-up period by subgroup.** a) Male b) Female.

deferral due to low Hb and the average number of deferrals per donor was higher for the reduced minimum interval strategies, but there was no evidence that donating blood more often led to measurable reductions in QoL, compared to donating every 12 weeks (for men) or every 16 weeks (for women). There were no differences in the self-reported fainting episodes, adverse events, or health care resource use across the randomised arms.

The results show that frequent donation of blood leads to Hb and non-Hb related deferrals, but the depletion of Hb and other self-reported symptoms does not have any detectable effect on QoL up to 2 years follow-up period in the INTERVAL trial. This finding is observed in even longer follow-up period of 4 years in the INTERVAL-extension study [12]. Our study adds that it is not only safe to collect blood more frequently than the current standard, but also a cost-effective strategy. Our study adds to the limited literature on the cost-effectiveness of alternative donation interval strategies for blood collection [3,31–40], and reported cost-

## a) Male

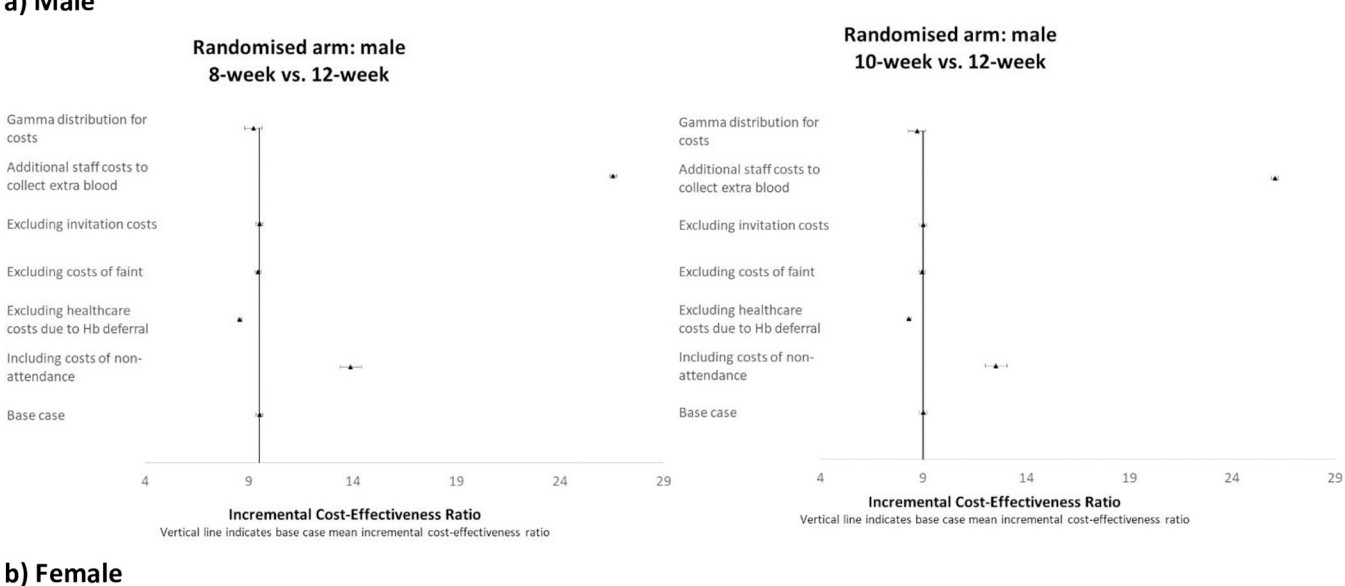

## b) Female

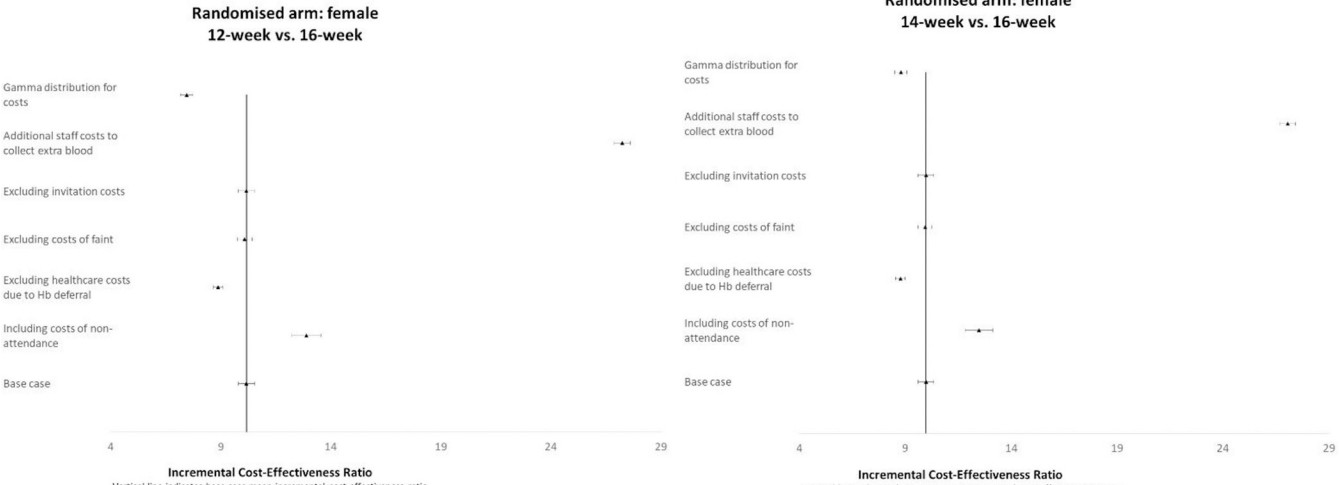

**Fig 4. Sensitivity analysis that reports the mean (95% CI) incremental cost-effectiveness ratios over two-year follow-up period according to alternative assumptions compared to the base case.** a) Male b) Female.

effectiveness results across subgroups of prime policy interest related to the blood service in England and other public-funded blood services.

Previous reports from the INTERVAL trial also showed that there was no difference in serious adverse events, cognitive function or levels of physical activity between people who gave blood most and least frequently [11]. However, a higher proportion of donors allocated to shorter donation intervals showed more self-reported symptoms including feeling more tired than usual, dizziness, feeling faint or more breathless, experiencing palpitation and symptoms compatible with restless legs syndrome in men and feeling more tired than usual, dizziness, feeling faint or more breathless in women [11]. On average, compared to people who gave blood less frequently, people who gave blood most frequently had lower iron and haemoglobin levels after two years.

A key strength of this CEA is that it was performed using donor-level data from a large, well-conducted RCT, with complete follow-up data for the main endpoints of interest, and included as a control arm, the current minimum donation interval in England. The large sample size allowed reporting both the overall effect of alternative minimum donation intervals on costs and outcomes and, also the effect according to subgroups of key policy relevance including donors whose blood is in high demand. By reporting cost-effectiveness results for these subgroups of key policy relevance, we extend a previous publication of the CEA that used the INTERVAL trial data for a more restriction range of donor subgroups, pre-specified for the original INTERVAL trial analysis [13].

The study has a few limitations. First, while the INTERVAL trial followed donors for up to four years, the higher Hb-related deferral rates in the reduced inter-donation interval arms could lead to a higher rate of donors leaving the blood donation registry in the long-run if the levels of Hb that were on average lower in the reduced interval arms after four years continue to fall and diverge subsequently. A similar concern is that we were not able to assess whether the increase in self-reported symptoms in the reduced interval arms led to more donors leaving the register over time [11]. Second, the RCT was undertaken at 25 static donor centres and therefore the cost-effectiveness results may not be generalisable to mobile sessions. Third, the CEA did not include the full range of costs that may differ across intervention groups. In particular, costs of non-attendance were excluded as data were not available on the number of non-attendances for each individual. In the sensitivity analysis, when these costs were approximated, the results show that the ICERs of the reduced interval strategies increased somewhat, but generally remained below an additional variable cost of £30 for an additional unit of blood donated. The results were most sensitive to the assumption that the static donor centres have sufficient capacity to collect the additional units of blood donated. This alternative assumption may not be realistic if reduced interval strategies are rolled-out to all donors attending static centres. However, if the reduced interval strategies are only applied to those groups whose blood type is in high demand, then current capacity (on average, 75%) may be sufficient to collect the additional units of blood at an incremental costs of no greater than £10 (the base case ICER). Fourth, we were unable to consider the additional costs that may be associated with the observed increase in self-reported symptoms in those giving blood more frequently, although there was no measurable reduction in QoL, physical activity or neurocognitive function in the those allocated to shorter intervals.

The study raises important questions for further research. First, the INTERVAL trial showed that on average, compared to people who gave blood less frequently, people who gave blood most frequently had lower iron and haemoglobin levels after four years and were more likely to have iron and haemoglobin levels below the minimum threshold required to donate blood. Evidence suggests that donors deferred for low Hb are much less likely to return for future donations than donors who are able to donate blood successfully [41]. Evidences from large national studies suggest that female and younger donors often have low level of ferritin store and their risk of ferritin depletion is relatively higher with reduced inter-donation interval [42,43]. Further research is warranted to customise donation intervals recognising that some donors, including those who self-report symptoms, could be at high risk of Hb and ferritin depletion and thereby more likely to stop donation. Further research on Hb and ferritin depletion and their consequent effect on costs and health outcome would be useful for informing sustainable donation strategies. Second, further studies could consider a wider set of interventions, including educational interventions for blood donation and investigate the relative impact of the wider set of interventions versus reducing inter-donation intervals, on the relative donation frequency and costs. Third, reducing the minimum donation interval is more cost-effective for older females, and those females whose blood groups is in high demand.

Finding effective ways to retain these donors is crucial. Blood collection agencies such as NHSBT should consider developing new retention strategies tailored to blood donors, taking into account the specific profiles of female/male donors including age, blood type, donation history, and ethnicity.

In summary, reducing the minimum donation interval yields additional units of whole-blood at a small additional cost over two years. The incremental costs per donation are relatively low for having inter-donation intervals that are shorter than current standard practice in the UK.

## Supporting information

**S1 Appendix.**
(DOCX)

## Acknowledgments

We acknowledge the role of colleagues from the Health Economics and Modelling of alternative blood donation strategies (HEMO) study, in particular Neil Hawkins, Silvia Perra, Jenny Turner, Crispin Wickenden, Catharina Koppitz, Gavin Cho and John Cairns. We also acknowledge other members of the INTERVAL Trial Group, in particular Matthew Walker, Jane Armitage, Willem Ouwehand, Nick Watkins, John Danesh and Simon Thompson.

## Author Contributions

**Conceptualization:** Zia Sadique, Sarah Willis, Kaat De Corte, Mark Pennington, Carmel Moore, Stephen Kaptoge, Emanuele Di Angelantonio, Gail Miflin, David J. Roberts, Richard Grieve.

**Data curation:** Zia Sadique.

**Formal analysis:** Zia Sadique.

**Investigation:** Zia Sadique, Sarah Willis, Kaat De Corte, Mark Pennington, Stephen Kaptoge, Emanuele Di Angelantonio, Gail Miflin, David J. Roberts, Richard Grieve.

**Methodology:** Zia Sadique, Richard Grieve.

**Project administration:** Carmel Moore.

**Validation:** Zia Sadique, Sarah Willis, Kaat De Corte, Mark Pennington, Carmel Moore, Stephen Kaptoge, Emanuele Di Angelantonio, Gail Miflin, David J. Roberts, Richard Grieve.

**Writing – original draft:** Zia Sadique, Sarah Willis, Kaat De Corte, Mark Pennington, Carmel Moore, Stephen Kaptoge, Emanuele Di Angelantonio, Gail Miflin, David J. Roberts, Richard Grieve.

**Writing – review & editing:** Zia Sadique, Sarah Willis, Kaat De Corte, Mark Pennington, Carmel Moore, Stephen Kaptoge, Emanuele Di Angelantonio, Gail Miflin, David J. Roberts, Richard Grieve.

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
