## [Decision Letter · Decision Letter 0]

22 Apr 2021

PONE-D-20-28101

Cost-effectiveness of alternative minimum recall intervals between whole blood donations

PLOS ONE

Dear Dr. Sadique,

Thank you for submitting your manuscript to PLOS ONE. After careful consideration, we feel that it has merit but does not fully meet PLOS ONE’s publication criteria as it currently stands. Therefore, we invite you to submit a revised version of the manuscript that addresses the points raised during the review process.

We look forward to receiving your revised manuscript.

Kind regards,

Bradford Dubik

Academic Editor

PLOS ONE

Journal Requirements:

4. Please provide additional details regarding participant consent. In the ethics statement in the Methods and online submission information, please ensure that you have specified what type you obtained (for instance, written or verbal, and if verbal, how it was documented and witnessed). If your study included minors, state whether you obtained consent from parents or guardians. If the need for consent was waived by the ethics committee, please include this information.

[Participants in the INTERVAL randomised controlled trial were recruited with the active collaboration of NHS Blood and Transplant England (www.nhsbt.nhs.uk), which has supported field work and other elements of the trial. DNA extraction and genotyping was co-funded by the National Institute for Health Research (NIHR), the NIHR BioResource (http://bioresource.nihr.ac.uk) and the NIHR [Cambridge Biomedical Research Centre at the Cambridge University Hospitals NHS Foundation Trust][*].The academic coordinating centre for INTERVAL was supported by core funding from: NIHR Blood and Transplant Research Unit in Donor Health and Genomics (NIHR BTRU-2014-10024), UK Medical Research Council (MR/L003120/1), British Heart Foundation (SP/09/002; RG/13/13/30194; RG/18/13/33946) and the NIHR [Cambridge Biomedical Research Centre at the Cambridge University Hospitals NHS Foundation Trust.]

 [No]

6. Thank you for submitting the above manuscript to PLOS ONE. During our internal evaluation of the manuscript, we found significant text overlap between your submission and the following previously published works.

- https://doi.org/10.3310/hsdr06400

- https://doi.org/10.1111/tme.12537

We would like to make you aware that copying extracts from previous publications, especially outside the methods section, word-for-word is unacceptable, even for works which you authored. In addition, the reproduction of text from published reports has implications for the copyright that may apply to the publications.

Please revise the manuscript to rephrase the duplicated text, cite your sources, and provide details as to how the current manuscript advances on previous work. Please note that further consideration is dependent on the submission of a manuscript that addresses these concerns about the overlap in text with published work.

Reviewers' comments:

Reviewer's Responses to Questions

**Comments to the Author**

1. Is the manuscript technically sound, and do the data support the conclusions?

Reviewer #1: Yes

Reviewer #2: Yes

2. Has the statistical analysis been performed appropriately and rigorously? 

Reviewer #1: Yes

Reviewer #2: Yes

3. Have the authors made all data underlying the findings in their manuscript fully available?

Reviewer #1: Yes

Reviewer #2: No

4. Is the manuscript presented in an intelligible fashion and written in standard English?

Reviewer #1: Yes

Reviewer #2: Yes

5. Review Comments to the Author

Reviewer #1: Blood donor recruitment is an important issue of the blood services worldwide and most blood services strongly depend on the repeat blood donors to keep their inventory. Thus, maintaining the repeat donor base is very important and as expected, is cost-effective, as demonstrated by the authors of this paper. In my opinion, although the cost-effectiveness in an important factor to be considered in the donor recruitment, blood donor safety should not be forgotten. In the recent years, many developed countries started investigating the serum ferritin as a surrogate of the iron stores of their donors, and most of them have found that although the hemoglobin (Hb) levels recover quite fast, this is not the case of ferritin. Based on this, they have implemented measures that include the extension of the allowed inter-donation period, and the deferral of donors with extremely low ferritin levels. At the Canadian Blood Services, based on a large national study of ferritin (Transfusion 2017; 57(3):564-570), the minimum hemoglobin level was increased to 130 g/L for male donors and the minimum inter-donation interval of female donors was changed to from 56 days to 84 days (four donations yearly). At Vitalant in the US, a strategy to defer teen donors with low ferritin from red blood cell donations (12 months for females, and 6 for males) and counsel them to take low‐dose iron for 60 days was implemented (Transfusion 2018; 58(12):2861-2867). At Sanquin in the Netherlands, a strategy to determine serum ferritin levels at every fifth donation, as well as in all first-time donors was implemented, and those with ferritin levels lower than 15ng/mL (WHO threshold) are deferred for 12 months, and those with values between 15 and 30ng/mL, for 6 months.

In the present manuscript, the authors mention that there was greater self-reporting of symptoms potentially related to blood donation, such as tiredness, feeling faint, breathlessness, dizziness, restless legs, and palpitations, especially among men, and that serum ferritin levels were especially lower among those donors allocated to the minimal inter-donation interval. They found that at the 2-year examination, the absolute decreases in mean hemoglobin concentrations were modest (around 1–2%), but they were large for serum ferritin (around 15–30%), reflecting the higher sensitivity of serum ferritin as an indicator of body iron stores compared to Hb. Deferral rates for low Hb were higher in the short interval arms, which suggests that in terms of serum ferritin, the deferral numbers should be higher, but usually donors do not report symptoms of low serum ferritin. Especially these papers reporting on the risks of low serum ferritin and policy changes in the blood services must be referred and discussed in more details to not give a false impression to the readers that the cost-effectiveness should prevail over donor safety, or that cost-effectiveness and donor safety must be balanced when deciding on the best donor recruitment strategies.

Also, it is shown that shortening the inter-donation interval of the repeat donors is cost-effective, the shorter interval strategies increasing average cost, with incremental cost-effectiveness ratios of ￡9.51 per additional whole-blood donation for the 8- vs 12-week interval for males, and ￡10.17 for the 12- vs 16-week interval arm for females. What would be the cost of recruiting new first-time or lapsed donors instead of shortening the inter-donation interval to achieve similar increase in whole blood donations?

I assume cases of vaso-vagal reaction (VVR) are included in fainting episodes. Were medications, including IV solutions, applied for the management of VVR? In case yes, were these costs included in the analysis?

It is interesting that the “high-demand” group and older female donors had the better cost-effectiveness, since those donors with rare blood group types are more conscious on the need of their blood, thus would be more cooperative to requests for blood donation. And older female donors are known to have higher Hb and ferritin levels, so they are more adapted for blood donation than younger female donors. Probably, educating these donors in the “high-demand” group and older females on the need of blood donation would contribute for the increase in the frequency of donations without need to reduce the inter-donation interval.

Minor points:

Page 6, Line 21: from the sentence“approved by the National Research Ethics Service approved”, remove the last “approved”

Page 7, Line 8: “Existing donor were defined” should be “Existing donors were defined”

Reviewer #2: PONE-D-20-28101: statistical review

SUMMARY. This study focuses on the relative cost-effectiveness of reduced intervals between consecutive blood donations, based on a large sample of blood donors randomly assigned to 12, 10 or 8 week (males), and 16, 14 or 12 week inter-donation intervals (females). The paper reads well and the statistical methods are correct. Results seem sound. The main results are clearly displayed in a battery of figures that are self-explaining. My only complaint is that some basic information is not included, see the points below. Such information is crucial for results reproducibility.

MINOR POINTS

1) page 10: "The analysis applied logistic regression

models (binary endpoints), linear regression models (continuous, univariate endpoints)". Please specify the dependent variables that have been used for each method

2) the statistical analysis relies on a battery logistic regression models, linear regression models and seemingly-unrelated regression models. However, the estimation results of these models are not included. Parameter estimates and standard errors (or p-values) should be made available within tables and described in the appendix. This is important for results replicability.

2) A GEE model is correctly exploited for modelling the SF-6D utility score. The Appendix should include a table with the estimates and the robust stardard errors. Again, this is for helping the interested reader who wants to replicate the results.

3) Page 11: "The confidence intervals around the ICER

were constructed by applying Taylor series expansion on the incremental estimates of cost and

volume of blood donated". Do the authors refer to a "delta method"? Please clarify.

TYPOS

1) Page 10 "The incremental analysis ... adjusted for ": I guess "was" is missing

2) Appendix figure 1: the label of the x-axis is "mean number of donation difference". I think that the word "mean" should be deleted

6. PLOS authors have the option to publish the peer review history of their article (what does this mean?). If published, this will include your full peer review and any attached files.

Reviewer #1: No

Reviewer #2: No

---

## [Author Response · Author response to Decision Letter 0]

19 Nov 2021

Responses to reviewers' and editor's comments are uploaded as separate files.

---

## [Decision Letter · Decision Letter 1]

28 Jul 2022

Cost-effectiveness of alternative minimum recall intervals between whole blood donations

PONE-D-20-28101R1

Dear Dr. Sadique,

We’re pleased to inform you that your manuscript has been judged scientifically suitable for publication and will be formally accepted for publication once it meets all outstanding technical requirements.

Kind regards,

Thomas Tischer

Staff Editor

PLOS ONE

Additional Editor Comments (optional):

Note from the journal office: It is not necessary to include the changes 1 - 3 suggested by the previous Academic Editor noted below as you addressed this in your reply to the journal office. However, please take care to address the typos mentioned.

I believe the paper was substantially improved, however, there is the risk of being considered a duplicate publication in case the issues below are not appropriately addressed:

1) It should be clearly stated that the second submission contains secondary analyses/results, and a sentence needs to be added in the manuscript acknowledging readers, peers and documenting agencies that the paper has been published in whole or in part in the HSDR, and the primary publication should be cited.

2) It is necessary to indicate in the title of the secondary publication that it is a secondary publication of the previous paper.

3) The permission from the previous journal is required for the new submission as well as to use the same figures and Tables.

I confirmed the authors stated in the text that this submission contains secondary analyses/results, but the title is quite similar to the previous publication, and there is no mention in the title that it is a secondary publication. Also, the authors need to get permission from HSDR to submit the subanalysis of the same data as well as to use the same figures and Tables.

Other minor revisions:

1. P7, L4: "lood type" should be "blood type"

2. P7, L5: period not required after illness

3. P11, L14: period required at the end of the sentence

4. P13, L9-10: "mean number of blood donation visits was relatively lower" should be "was relatively higher"? Please check

5. P16, L9: "noevidence" should be "no evidence"

I disclose that I participated as a reviewer for the initial evaluation of this manuscript.

Reviewers' comments:

Reviewer's Responses to Questions

**Comments to the Author**

1. If the authors have adequately addressed your comments raised in a previous round of review and you feel that this manuscript is now acceptable for publication, you may indicate that here to bypass the “Comments to the Author” section, enter your conflict of interest statement in the “Confidential to Editor” section, and submit your "Accept" recommendation.

Reviewer #2: All comments have been addressed

2. Is the manuscript technically sound, and do the data support the conclusions?

Reviewer #2: (No Response)

3. Has the statistical analysis been performed appropriately and rigorously? 

Reviewer #2: (No Response)

4. Have the authors made all data underlying the findings in their manuscript fully available?

Reviewer #2: (No Response)

5. Is the manuscript presented in an intelligible fashion and written in standard English?

Reviewer #2: (No Response)

6. Review Comments to the Author

Reviewer #2: (No Response)

7. PLOS authors have the option to publish the peer review history of their article (what does this mean?). If published, this will include your full peer review and any attached files.

Reviewer #2: No

---

## [Editor Report · Acceptance letter]

9 Aug 2022

PONE-D-20-28101R1 

Cost-effectiveness of alternative minimum recall intervals between whole blood donations 

Dear Dr. Sadique:

I'm pleased to inform you that your manuscript has been deemed suitable for publication in PLOS ONE. Congratulations! Your manuscript is now with our production department. 

Kind regards, 

on behalf of

Dr. Thomas Tischer 

Staff Editor

PLOS ONE